# Adaptive and Innate Cytotoxic Effectors in Chronic Lymphocytic Leukaemia (CLL) Subjects with Stable Disease

**DOI:** 10.3390/ijms24119596

**Published:** 2023-05-31

**Authors:** Valentina Rubino, Flavia Carriero, Anna Teresa Palatucci, Angela Giovazzino, Stefania Leone, Valerio Nicolella, Martina Calabrò, Rosangela Montanaro, Vincenzo Brancaleone, Fabrizio Pane, Federico Chiurazzi, Giuseppina Ruggiero, Giuseppe Terrazzano

**Affiliations:** 1Department of Translational Medical Sciences, University of Naples “Federico II”, 80131 Naples, Italy; valentina.rubino@unina.it (V.R.); angela.giov@tiscali.it (A.G.); valerio.nicolella@mensa.it (V.N.); 2Department of Science, University of Basilicata, 85100 Potenza, Italy; flavia.carriero@unibas.it (F.C.); anna.palatucci@unibas.it (A.T.P.); rosangela.montanaro@unibas.it (R.M.); vincenzo.brancaleone@unibas.it (V.B.); 3Division of Hematology, Department of Clinical Medicine and Surgery, University of Naples “Federico II”, 80131 Naples, Italy; stefania0leone@gmail.com (S.L.); martina.calabro@hotmail.it (M.C.); fabrizio.pane@unina.it (F.P.); ambulatoriochiurazzi@unina.it (F.C.)

**Keywords:** chronic lymphocytic leukaemia, HLA class I molecules, cytotoxic T cells, NK cells, immune activation profile

## Abstract

Chronic lymphocytic leukaemia (CLL) is characterised by the expansion of a neoplastic mature B cell clone. CLL clinical outcome is very heterogeneous, with some subjects never requiring therapy and some showing an aggressive disease. Genetic and epigenetic alterations and pro-inflammatory microenvironment influence CLL progression and prognosis. The involvement of immune-mediated mechanisms in CLL control needs to be investigated. We analyse the activation profile of innate and adaptive cytotoxic immune effectors in a cohort of 26 CLL patients with stable disease, as key elements for immune-mediated control of cancer progression. We observed an increase in CD54 expression and interferon (IFN)-γ production by cytotoxic T cells (CTL). CTL ability to recognise tumour-targets depends on human leukocyte antigens (HLA)-class I expression. We observed a decreased expression of HLA-A and HLA-BC on B cells of CLL subjects, associated with a significant reduction in intracellular calnexin that is relevant for HLA surface expression. Natural killer (NK) cells and CTL from CLL subjects show an increased expression of the activating receptor KIR2DS2 and a reduction of 3DL1 and NKG2A inhibiting molecules. Therefore, an activation profile characterises CTL and NK cells of CLL subjects with stable disease. This profile is conceivable with the functional involvement of cytotoxic effectors in CLL control.

## 1. Introduction

Chronic lymphocytic leukaemia (CLL), the most common leukaemia in adulthood, is characterised by the expansion of mature B cells expressing CD5, CD19 and CD23 [1,2,3]. The clinical outcome of CLL is very heterogeneous. In this context, early stage asymptomatic patients show partial long-term benefits or no therapeutic effect from early drug treatment, while patients with advanced disease necessarily require first-line therapy to limit disease progression [1,2,3]. Several alterations, such as p53 mutation, immunoglobulin heavy-chain variable gene (IGHV) mutational status, cytogenetic and epigenetic modification [4,5,6], as well as micro-environmental stimulation [7], have been proposed as predictive and/or diagnostic biomarkers of the disease. However, the need for additional disease markers is recognised as essential to improve the diagnostic and predictive criteria of CLL disease. Regarding therapies, the recent success of immunotherapy in refractory/relapsed CLL [8] highlights the possibility of effectively manipulating immune effectors to target leukaemia clones.

In this context, the study of the immune profile of subjects characterised by clinically stable CLL could provide interesting data on the possible balance between the immune response and disease progression. Although the mechanisms underlying the immune-mediated control of tumour cells in human models are still largely undefined, the involvement of natural killer cells (NK), cytotoxic CD8^+^ T lymphocytes (CTL) and Interferon (IFN)-γ dependent pathways in the control of tumour progression have been widely recognised [9,10,11,12,13,14,15,16,17,18].

The presence of a functional T-cell compartment and the occurrence of multiple defects, such as impaired proliferation, cytotoxicity and inability to form effective immune-synapses, have been reported in subjects with CLL [19,20,21,22]. Moreover, the analysis of NK cell activity revealed defects in lysis, including antibody-dependent cellular cytotoxicity (ADCC), but also unaffected NK functions in CLL [23,24,25,26,27]. Therefore, the role of these innate cytotoxic effectors and the involvement of their receptor repertoire need to be further characterised in CLL.

In this respect, the ability of NK effectors to recognise and kill susceptible cell targets depends on a complex signalling network involving a recognition repertoire, which physiologically includes both activating and inhibiting of NK receptors [28,29,30]. T-cell activation requires the recognition of antigenic peptides within human leukocyte antigen (HLA)-I molecules expressed on the membrane of target cells [31,32]. The existence of an association between the tumour progression and the selection of HLA-I molecules defective clones has been described in several cancers [33,34,35]. In this context, the possibility that altered HLA-I expression might affect single allele/loci of the HLA complex has been largely proposed [33,34,35,36]. Antigen presentation via HLA-I molecules depends on a complex intracellular molecular network of the antigen processing machinery (APM), physiologically involved in the correct assembly of the HLA-I molecular complex and in the loading of the peptide into the HLA-I groove [36,37,38]. The involvement of HLA-I molecules in the immune escape mechanisms of CLL appears to correlate with a prevalent downregulation of HLA-C molecules, as well as a defective expression of HLA-A and HLA-B antigens [39]. Furthermore, in very advanced CLL requiring transplantation, individuals homozygous for one or more HLA-I loci have a worse prognosis than heterozygotes [40]. Therefore, a more complete evaluation of both the expression of HLA-I molecules and the correct functional architecture of the APM need to be further investigated in the CCL, as key elements for immune-mediated control of leukaemia progression.

Immune response is a complex phenomenon aimed to ensure protection against pathogens, also maintaining tissue homeostasis [41,42]. Cell-mediated control of immune effectors are usually dependent on a heterogeneous group of differentiated cell subsets including the FoxP3^+^ CD4^+^ CD25^+^ regulatory T cells (Treg) [43,44,45,46]), the interleukin (IL)-10 producing T_R1_ [47,48], the transforming growth factor (TGF)-β producing T_H3_ [49] lymphocytes.

We recently described that T_R3-56_, co-expressing CD3 and CD56 molecules, represents a peculiar human regulatory T-cell subset, phenotypically, metabolically and genetically distinct from NKT subset [50,51]. T_R3-56_ subset is preferentially involved in the control of cytotoxic activity and in the production of IFN-γ by cytotoxic T cells [5,51]. T_R3-56_ cells are significantly reduced in type 1 diabetes (T1D) at disease onset [51] and are inversely correlated with the presence of activated cytotoxic T lymphocytes with a skewed Vβ T-cell repertoire in the bone marrow (BM) of myelodysplastic subjects [52].

Here, we address the analysis of cytotoxic effectors, belonging to both innate and adaptive compartment, as well as of HLA-I expression by B cell clones, in CLL subjects with stable disease.

## 2. Results

### 2.1. Circulating T Cells, NK, Treg and T_R3-56_ in CLL Patients

We analysed the number and percentage of circulating T and NK lymphocytes in CLL subjects showing stable disease, as compared with the sex-age matched healthy subject group.

According to the literature [19,20,21,22,23,24,25,26,27], we observed that the reduction in T (18.54 ± 2.45 vs. 74.64 ± 0.67 in controls; *p* < 0.0001) and NK (3.90 ± 0.58 vs. 9.48 ± 0.45 in controls; *p* < 0.0001) lymphocyte percentages (Figure 1A,C, respectively) is accompanied by an increase in the absolute number of T (2268 ± 225.4 vs. 1456 ± 38.58 in controls; *p* < 0.0001) and NK cells (379.7 ± 44.94 vs. 197.5 ± 5.23 in controls; *p* < 0.0001) in CLL subjects (Figure 1B,D, respectively). Furthermore, we observed a significant decrease in the CD4/CD8 T-cell ratio in CLL patients when compared to healthy subjects (2.21 ± 0.42 vs. 2.53 ± 0.09 in controls; *p* < 0.001) (Figure 1E).

In addition, we evaluated the presence of both Treg [43,44,45] and T_R3-56_ [51,52] in the cohort of CCL subjects. As shown, Treg percentage is reduced in CLL subjects when compared to healthy controls (1.31 ± 0.17 vs. 2.43 ± 0.18 in healthy controls; *p* < 0.0001) (Figure 2A,B), while the Treg absolute number (Figure 2B) is increased (177.2 ± 25.8 vs. 21.17 ± 1.01 in healthy controls; *p* < 0.0001). In contrast, the percentage of the T_R3-56_ population shows no significant difference in the two groups (3.11 ± 0.47 vs. 3.91 ± 0.39 in healthy controls) (Figure 2C). In addition, a greater number of circulating T_R3-56_ cells was observed in subjects with CLL (341.5 ± 61.99 vs. 91.31 ± 11.58 in healthy controls; *p* < 0.001) (Figure 2D). Moreover, the level of circulating Treg and T_R3-56_ cells directly correlates to the CLL cohort (Spearman r = 0.625; *p* < 0.005) (Figure 2E).

To describe the T-cell profile of our CLL cohort, we analysed the percentage of Treg and T_R3-56_ subset in the T-cell population. In this regard, an increased percentage of Tregs (13.81 ± 1.07 vs. 7.68 ± 0.53 in healthy controls; *p* < 0.0001) and T_R3-56_ (15.48 ± 2.34 vs. 6.01 ± 0.63 in healthy controls; *p* < 0.0001) was observed in the T-cell subset from CLL subjects (Figure 3A,B, respectively).

Therefore, an increase in regulatory T-cell subsets appears to characterise the T-cell compartment in CLL patients, as already highlighted [53,54]. This evidence suggests that an increased rate of Treg and T_R3-56_ differentiation within the T-cell compartment characterises the CLL individuals with stable disease.

### 2.2. Circulating Cytotoxic T Cells Are Characterised by an Increased Expression of CD54 and High Production of Interferon-γ in CLL Subjects with Stable Disease

We analysed the CD54 expression, as a marker of antigen-dependent CTL activation [52,55,56,57], and the production of IFN-γ by T cells and NK effectors in CLL subjects.

An increased expression of CD54 (13.26 ± 0.80 vs. 6.72 ± 1.39 in healthy controls; *p* < 0.0001) and a higher IFN-γ production (62.94 ± 5.97 vs. 34.84± 4.81 in healthy controls; *p* < 0.005) by CTL characterise CLL subjects (Figure 4A,C, respectively).

Furthermore, CD3^+^ CD4^+^ T lymphocytes present a higher CD54 expression (7.04 ± 0.25 vs. 2.51 ± 0.5 in controls; *p* < 0.0001) (Figure 4B,D), while the IFN-γ production by CD3^+^ CD4^+^ cells are substantially comparable between CLL and healthy controls (26.06 ± 4.50 vs. 20.65 ± 2.97, respectively). Similar results were observed when IFN-γ was analysed in the NK population (19.95 ± 4.21 vs. 24.02 ± 4.34 in healthy controls) (Figure 4E).

Taken in all, CLL subjects with stable disease appear to be characterised by an activation profile of circulating CTL.

### 2.3. A Lower Expression of HLA-A and HLA-BC Molecules and a Reduction of Intracellular Calnexin Characterise Circulating B Cells, but Not T Lymphocytes from CLL Subjects with Stable Disease

To evaluate the expression of HLA-I molecules, we used specific monoclonal antibodies (mAbs) able to recognise specific HLA-I molecular structures [58,59,60,61,62] (see Materials and Methods section). Analysis of HLA-I expression on circulating B lymphocytes of CLL subjects with stable disease revealed a significant reduction in HLA-A (37.49 ± 5.77 vs. 104.9 ± 27.81 in healthy controls; *p* < 0.005), HLA-BC (47.58 ± 7.25 vs. 128.4 ± 38.57 in controls; *p* < 0.05) and β2-microglobulin surface level (54.29 ± 8.96 vs. 156.81 ± 41.91 in healthy controls; *p* < 0.005) (Figure 5A). The expression of the HLA-ABC molecules, evaluated as a whole, on the membrane of the leukaemia B-cell clones was not significantly different from the B lymphocytes of the control population (61.32 ± 9.31 vs. 110.7 ± 32.78 in healthy controls) (Figure 5A). At variance, HLA-I expression on T lymphocytes showed no significant difference between CLL subjects and healthy controls (Figure 5B).

To investigate the molecular basis of the observed altered expression of HLA-I on leukaemia clones, we analysed the expression of the APM molecules in circulating B and T lymphocytes from CLL subjects. A reduced level of calnexin (10.25 ± 1.63 vs. 20.42 ± 3.97 in healthy controls) was found to characterise the B cell compartment from CLL subjects (Figure 6A). In addition, TAP-1 [36,37,38,61] (2.47 ± 0.32 vs. 0.71 ± 0.29 in controls), Tapasin [36,37,38,59] (2.27 ± 0.25 vs. 0.99 ± 0.32 in healthy controls), LPM7 (3.81 ± 0.72 vs. 1.5 ± 0.44 in healthy controls) and LMP10 (2.4 ± 0.27 vs. 0.73 ± 0.31 in healthy controls) molecules increased in the T-cell compartment (Figure 6B).

Extracellular expression of calreticulin represents a marker of endoplasmic reticulum (ER) stress, usually associated with intracellular protein misfolding and immunogenic cell death processes [63]. The extracellular calreticulin expression increased in both circulating B (30.67 ± 4.40 vs. 13.74 ± 3.76 in healthy controls) and T lymphocytes (1.5 ± 0.22 vs. 0.71 ± 0.35 in healthy controls) of the CLL subjects (Figure 7).

In addition, the binding of the specific anti HLA-ABC mAb revealed an increase in HLA-I surface expression when B cells and T cells from CLL subjects were cultured with IFN α-2b (Figure 8A,B). This finding suggests that the defective expression of HLA-I molecules in CLL B cell clones might be susceptible to cytokine-mediated in vitro restoration.

### 2.4. Circulating NK Lymphocytes from CLL Subjects with Stable Disease Show Higher Expression of KIR2DS2 Activating Receptor and Significant Reduction of 3DL1 and NKG2A Inhibiting Molecules

We analysed the expression of both activating and inhibiting receptors on NK effectors of CLL subjects with stable disease. Increased expression of the activating receptor KIR2S2 (CD158j) [64] in CLL patients compared to healthy donors was observed (6.10 ± 2.01 vs. 0.10 ± 0.12, respectively; *p* < 0.001) (Figure 9A). No differences were found in the expression of the activating molecules Nkp46 (CD335), Nkp30 (CD337), Nkp44 (CD336) and NKG2D [27,29,30] (Figure 9A).The expression of the activating receptor 2B4 (CD244) [65,66] is decreased in CLL subjects (86.63 ± 4.82 vs. 99.77 ± 0.11 in healthy controls; *p* < 0.0005) (Figure 9A).

Analysis of the inhibitory receptors 3DL1 (CD158e1/e2) and CD85k (ILT3), which binding to HLA-G molecules mediates NK inhibition [67,68,69], revealed reduced levels of the 3DL1 (12.63 ± 3.05 vs. 21.12 ± 4.10 in healthy controls; *p* < 0.05) associated with increased expression of CD85k (1.60 ± 0.29 vs. 0.41 ± 0.09 in healthy controls; *p* < 0.01) (Figure 9A).

Intriguingly, an increase in HLA-G was detected in circulating B cells (26.07 ± 3.78 vs 10.03 ± 0.99 in healthy controls; *p* < 0.005) and T cells (1.41 ± 0.2 vs. 0.8 ± 0.09 in healthy controls; *p* < 0.05) from subjects with CLL (Figure 9B,C).

An inverse correlation was observed between the expression of HLA-G on B lymphocytes and the level of CD85k, the specific HLA-G receptor, on NK effectors of CLL individuals (Spearman r = 0.522; *p* < 0.05) (Figure 9D).

Expression of the NK receptors on the CTL has been extensively described [70,71,72]. In CLL subjects, analysis of the CTL receptor repertoire revealed an increased KIR2S2 (CD158j) expression (3.66 ± 1.12 vs. 0.65 ± 0.29 in healthy controls), while no difference was observed for the other receptors (CD335, CD336, CD337, NKG2D, CD244, CD158e1/e2 and CD85k) (Figure 10A). No significant correlation was revealed between HLA-G expression on circulating B cells from the CLL subjects and CD85k expression on peripheral CTL, although a negative correlation trend has been observed (Spearman r= −0.265) (Figure 10B).

Altered HLA-E molecule expression by B lymphocytes has been described in CLL [73,74]. Our data confirm such observation in CLL circulating B cells (15.24 ± 1.80 vs. 3.31 ± 1.27 in healthy controls), while no significant difference was revealed in the HLA-E expression by T cells of the CLL subjects with respect to controls (1.36 ± 0.19 vs. 0.79 ± 0.18 in healthy controls) (Figure 11A,B, respectively).

HLA-E molecule binding with NK effectors has been described to generate key inhibitory signals, mediated by the NKG2A receptor as well as activation of the NK cells after binding with the NKG2C molecule [75,76,77]. We observed a reduced NKG2A expression on NK cells of CLL subjects (36.63 ± 3.41 vs. 54.12 ± 3.45 in healthy controls; *p* < 0.01), while no difference was revealed in the level of NKG2C expression between CLL subjects and healthy individuals (Figure 11C). No correlation was observed between NKG2A and NKG2C expression on NK cells and HLA-E expression on CLL B lymphocytes (Figure 11D,E, respectively). In this context, a trend of negative association was revealed (Spearman r = 0.187 for NKG2A/HLA-E; Spearman r = −0.254 for NKG2C/HLA-E) (Figure 11D,E, respectively).

A decrease in NKG2A was observed in the CTL from CLL subjects (2.42 ± 0.35 vs. 7.63 ± 2.11 in healthy controls; *p* < 0.005), while no difference was revealed in NKG2C expression (Figure 12A).

## 3. Discussion

Here, we describe that the reduced expression of HLA-A and HLA-BC Class-I molecules on B lymphocytes is accompanied by the expression of an activation profile by both adaptive and innate cytotoxic effectors in CLL subjects with stable disease. Indeed, we observed an increased CD54 expression, a greater IFN-γ production by CTL and the presence of a NK receptor repertoire with preferential expression of activating molecules in our cohort of CLL patients.

To investigate on the role of immune-mediated pathways in the control of CLL progression, we analysed the immune profile of the CLL subjects with stable disease, as compared with a cohort of 20 sex/age matched healthy controls. In this model, we focused on cytotoxic effectors as well as IFN-γ dependent pathways, largely described to play a key role in the control of cancer progression [9,10,11,12,13,14,15,16,17,18].

We confirmed the already described reduction of T and NK percentage, with numerical increase in both cell types in patients affected by CLL [19,20,21,22,23,24,78,79,80]. In addition, we also observed the decreased CD4/CD8 T-cell ratio [19,20,21,22,78,79,80] in our CLL cohort. These data suggest that a higher proportion of circulating CTL characterises the T-cell compartment in CLL subjects with stable disease. The functions of immune effectors have been described to depend on cell-mediated regulatory networks involving Treg [41,42,43,44,45] and T_R3-56_-cell subset that we recently described to preferentially modulate CTL effectiveness in autoimmune [51] and in tumour disease [52,81,82].

Our data indicate that the Treg and T_R3-56_ percentages decreased when evaluated on total lymphocytes, but increased when they are specifically analysed in the T-cells compartment alone. Furthermore, the absolute number of circulating Treg and T_R3-56_ is significantly higher in CLL patients when compared to healthy controls. Since lymphocytes are mainly composed of B cells in CLL patients, the small percentage of T cells within the lymphocyte compartment appears to exhibit a preferential expansion of Treg and T_R3-56_ regulatory cell subsets, as a possible immune escape mechanism. Our data confirm the Treg expansion in CLL [78,79,80] and highlight an expansion also of the recently characterised regulatory T_R3-56_ population [51,52,81,82].

In our CLL cohort, we observed the increased CD54 expression, largely associated with antigen-dependent T-cell activation [52,55,56,57], together with increased IFN-γ production by CTL. Such a profile is conceivable with the involvement of adaptive cytotoxic effectors in the control of leukemic B cells, potentially fostering the establishment of a stable CLL disease [16,17,19,20,78,79,80].

The expression of HLA-I, required for CTL-dependent recognition of antigens, has been extensively related to the immune-mediated control of cancer initiation/progression, as well as to the efficacy of immunotherapy in mouse and human models [32,33,34,35,36,37,38]. In this regard, the altered expression of HLA-I has been described to have a relevant role in the immuno-editing processes of the tumour context [34,35,36,37,38]. Modulation of the expression of individual HLA-I loci/alleles has been implicated in determining defective CTL activation [34,35,36,37,38,39].

In this article, we describe a reduced expression of HLA-A and HLA-BC molecules on B cells from CLL subjects with stable disease, without changes in the T-cell compartment. To investigate the mechanisms underlying HLA-I altered expression, we analysed the APM molecules [36,37,38], essential for the correct assembly of the HLA-I. We observed a reduced surface expression of HLA-A and -BC molecules, associated with defective intracellular calnexin levels in B lymphocytes from CLL individuals. Conversely, we found that the unaffected surface HLA-I expression is accompanied by an increase in intracellular TAP-1, Tapasin, LMP7 and LMP10 APM molecules in the T-cell compartment. Such features are conceivable with the hypothesis of an ongoing selection process mediated by activated CTL, able to preferentially target tumoral B-cell compartment, as largely proposed [37,38]. Based on our data, we can speculate that the increased IFN-γ production by CTL may induce APM molecules in circulating T lymphocytes of CLL individuals, also affecting surface calreticulin expression, usually associated with the occurrence of an overloading of the intracellular protein folding by B and T cells, as well as with immunogenic cell death processes [63]. The lack of HLA-I expression by cancer cells has been associated with the inability to respond to the immune therapeutic approaches [36,37,38]. In this context, multiple attempts to restore HLA-I expression in clinical settings have been proposed [36,37,38]. Here, we demonstrated that in vitro IFN α-2b treatment of B and T cells, obtained from individuals with stable CLL, increased their HLA-I surface expression. Therefore, it is conceivable that leukemic B cells in our cohort of CLL subjects with stable disease are potentially susceptible to cytokine-mediated HLA-I upregulation, as described [83].

Overall, our data suggest that the presence of activated CTL might be involved in the control of the expansion of transformed B-cell clones, as well as in the modulation of HLA-I expression on B lymphocytes in CLL subjects with stable disease. However, in order to demonstrate the relevance of this hypothesis in CLL subjects, further studies evidencing similarities and/or differences between subjects with stable and advanced-stage CLL regarding the functional role of both the CD8 or CD4 expressing T_R5-56_ lymphocytes will be needed [50,51] (see also Study Limitations).

Moreover, we analysed the NK receptor repertoire expressed by circulating NK lymphocytes [27,28,29,30] and by CTL of CLL subjects [70,71,72]. In this regard, we observed a significant up-regulation of the activating receptor KIR2S2 [64] in NK cells from CLL subjects, without significant changes in the expression of other activating molecules CD335, CD336, CD337, NKG2D and NKG2C [27,28,29,30,77]. We highlighted a reduced expression of CD244, already demonstrated to participate in activation processes in human NK models [65,66]. Concerning the inhibitory receptor repertoire of NK effectors in the CLL cohort, a significant down-modulation of 3DL1 and NKG2A [75,76,77] inhibitory molecules was revealed, while an increased level of the HLA-G binding CD85k receptor was observed [67,68]. Furthermore, we described that NK-dependent IFN-γ production was not significantly different from healthy controls. In addition, we found an increase in the activating KIR2S2 [64], accompanied by a reduction of the inhibitory NKG2A molecule [75,76,77] on CTL, in our cohort of CLL patients.

The role of altered HLA-G [78,80,81] expression in CLL is still controversial and of unclear prognostic significance [81]. Here, we demonstrated a preferentially increased expression of HLA-G on B cells, but not on T cells, from CLL patients with stable disease. Furthermore, we described a correlation between HLA-G expression by B cells and the percentage of the HLA-G binding inhibitory receptor CD85k on NK effectors in CLL subjects, while no significant correlation between HLA-G on B cells and CD85k expression by T lymphocytes was observed. The ability of HLA-G to induce CD85k expression on lymphoid human cell lines has been also observed [69]. Altered HLA-E molecule expression by B lymphocytes has been described in CLL [26] and is confirmed in our cohort. In this regard, no significant correlation was observed between B lymphocyte HLA-E expression and the HLA-E binding molecules NKG2A and NKG2C in the NK compartment or CTL compartment of subjects with CLL.

Taken together, our data suggest that circulating CTLs exhibit a potential activation profile involving TCR and NK-receptor-dependent pathways in CLL individuals with stable disease. In this regard, a more complex profile was observed in NK cells. Indeed, a significant up-regulation of the activating receptor KIRDS2, with a decrease in the level of the 3DL1 and NKG2A inhibitory molecules, is accompanied by both a reduced expression of the activating molecule CD244 and an increasing amount of the inhibitory receptor CD85k. Alterations of HLA class I molecules, as well as HLA-G and HLA-E, on B cells are of potential interest, as they could represent a putative escape mechanism of neoplastic clone B from CTL and NK recognition in CLL patients with stable disease. However, since the balance between the control of CTL and NK effectors may be crucial for disease determinism and outcomes, further studies of the functional mechanisms of the immune response and its regulation in CLL disease are needed (see Study Limitations).

## 4. Materials and Methods

### 4.1. Patients and Controls

Twenty-six patients, diagnosed as stage 0–1 LLC, according to *Rai* system, and as stage A, according to *Binet* system [84], all belonging to the Low Risk category, according to the *CLL-IPI* score [85], were enrolled in the study. Detailed description of clinical characteristic of our cohort is reported in Table 1.

Twenthy healthy donors, sex/age matched with CLL subjects, were also enrolled in the study.

Informed consent was obtained from each individual before each sample collection. Study was approved by the local Ethical Committee (protocol n. 347/19). None of the patients recruited was receiving immune-modifying medical treatments. Enrolled patients were not affected by immune-mediated diseases and acute or chronic viral infections.

### 4.2. Immune-Fluorescence and Flow Cytometry

FITC, PE, PEcy5, PEcy7 and APC labelled anti-CD3, -CD4, -CD8, -CD56, -CD25, -CD45, -CD54, -NKG2D, -HLA-E, -HLA-G and control isotype-matched monoclonal antibodies (mAbs) were purchased from BD PharMingen (San Jose, CA). FITC and PE-labelled anti-CD158j, -CD158e1/e2, -CD85, -nkp44, -CD244, -CD335, -CD337 mAbs were purchased from Beckman-Coulter, Paris, France. PE-labelled anti-NKG2A, -NKG2C mAbs were from R&D Systems, Inc. (Minneapolis, MN, USA). The TP 25.99.8.4 (anti-HLA-A, -HLA-B and -HLA-C associated with β2m) [58], LGIII-147.4.1 (anti-HLA-A associated with β2m) [59], B1.23.1 (anti-HLA-B and -C associated with β2m) [60], L368 (anti- β2m) [61], NOB-1 (anti-TAP1) [58] NOB-2 (anti- TAP2) [61], TO-5 (anti-calnexin) [62], TO-11 (anti-calreticulin) [62], TO-3 (anti-tapasin) [62], TO-7 (anti-LMP10) [61], Sy-3 (anti-LMP7) [63] mAbs were donated by Prof. Soldano Ferrone from Massachusetts General Hospital USA, and have been largely described for the characterisation of HLA and APM molecules in human tumours [36,37,38]. To analyse Foxp3 expression, intracellular staining was performed by using the anti-human Foxp3 kit (eBioscience, San Diego, CA, USA), following the manufacturer’s instructions. Anti-human interferon (IFN)-γ and isotype-matched mAbs were purchased from Becton Dickinson PharMingen, San Jose, CA, USA. The production of IFN-γ has been analysed by culturing purified peripheral blood mononuclear cells (PBMC) overnight in the presence of PMA and Ionomycin (Sigma-Aldrich, St. Louis, MO, USA). To avoid extracellular cytokine export, the cultures were incubated in the presence of 5 μg/mL of Brefeldin-A (Sigma-Aldrich), as described [86]. Intracellular staining with the specific mAbs was performed by a fixing/permeabilisation kit (Caltag, Burlingame, CA, USA), following the manufacturer’s instructions. INF α-2b was from Sigma-Aldrich.

T_R3-56_ lymphocytes have been identified by the co-staining with anti-human CD3 and anti-human CD56 mAb, as described [51,52]. All phenotypes referred to flow cytometry analysis of the lymphocyte population gated by using *Forward Scatter* (FSC) and *Side Scatter* (SSC) parameters, as well as CD45 labelling.

Flow cytometry and data analysis were performed by a two-laser equipped FACScalibur apparatus and the CellQuest analysis software (Becton Dickinson). Flow cytometry gating strategy is reported in Figure 13.

For the comparative analysis of CD54 expression on T lymphocytes as well as of HLA-I and APM molecule expression in B and T cells, immune-fluorescence data were expressed as ratio of mean intensity fluorescence (MIF) value for each lymphocyte subset and the control MIF value obtained after staining the same cell population with the isotype control mAb, as described [5,52].

### 4.3. Statistical Analysis

Statistical evaluation of data, by *InStat 3.0* software (GraphPad Software Inc., San Diego, CA, USA), was performed by *Mann–Whitney*, *Wilcoxon matched-pairs signed rank test* or *Spearman’s correlation test*, as indicated. Two-sided *p* values less than 0.05 were considered significant.

## 5. Conclusions

Our observations are conceivable with the hypothesis that the functional efficacy of innate and adaptive cytotoxic immune effectors may participate in the immunological scenario underlying disease stability in chronic lymphocytic leukaemia (CLL). Indeed, we described an overactivated profile of adaptive cytotoxic T lymphocytes (CTL), while the receptor-activating natural killer (NK) lymphocyte repertoire was substantially maintained in a cohort of CLL subjects with stable disease.

Stable CLL could be considered as a peculiar case of *a long-term equilibrium* between the clonally transformed B-cell population and immune-mediated antineoplastic activity. Our data suggest the potential role of some of the immune mechanisms underlying this unique condition. However, further functional studies are needed to understand whether these mechanisms are actually present in CLL and to explore on the complex scenario underlying CLL pathogenesis/progression (see Study Limitations). Characterisation of the complex interaction between CLL leukaemia cells and immune effectors is extremely useful not only for a better understanding of CLL pathophysiology, but could add key information for designing innovative immunotherapeutic strategies.

### Study Limitations

This research is based on the immunophenotypic characterisation of immune effectors and molecules involved in antigen recognition and not on functional analysis. The present study only includes observations on patients with stable CLL and not patients with unstable and/or advanced CLL, as the latter enrolled cohorts that were too small and did not allow for robust statistical analysis. Therefore, future studies are needed to demonstrate functional correlates between immune response regulation and its impact on CLL disease determinism, including stable and unstable CLL patients.

## Figures and Tables

**Figure 1 ijms-24-09596-f001:**
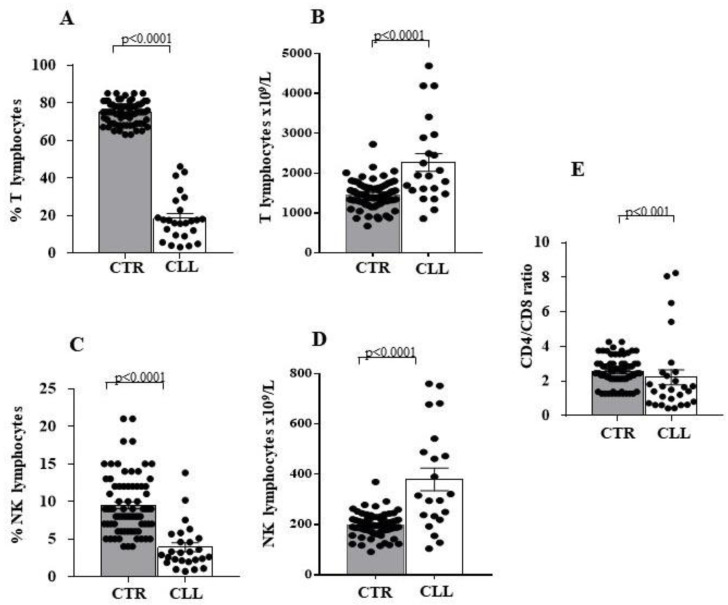
Significant decrease in the percentage, associated with increased number characterises circulating T and natural killer (NK) cells of chronic lymphocytic leukaemia (CLL) subjects with stable disease. Panels (**A**,**C**) indicate the percentage of circulating T and NK lymphocytes; panels (**B**,**D**) indicate the number of T and NK cells in peripheral blood; panel (**E**) shows the CD4/CD8 T-cell ratio of circulating lymphocytes. Comparative analysis of LLC and healthy controls shows decreased percentage of circulating T and NK effectors associated with a significant increase in their number. Grey and white columns indicate data obtained in healthy controls (CTR in *x* axis) and CLL individuals (CLL in *x* axis), respectively. Statistical evaluation of data has been performed by means of the *Mann–Whitney* test. Statistical significance values are indicated. The applied flow cytometry gating strategy is reported in Section 4.2.

**Figure 2 ijms-24-09596-f002:**
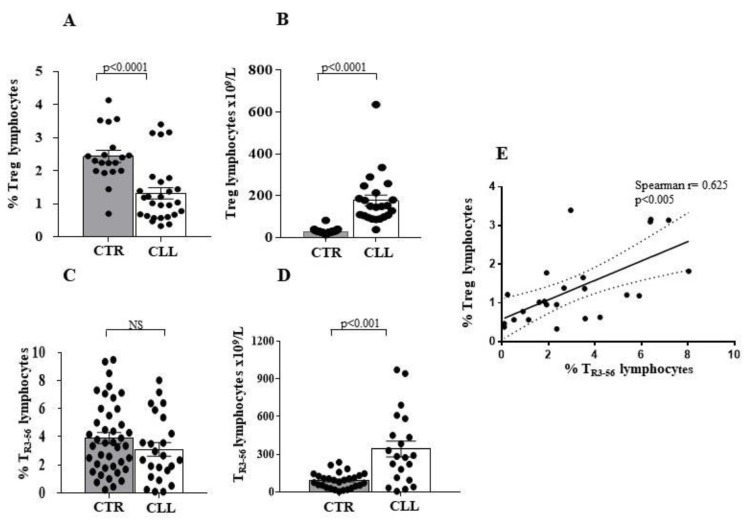
Analysis of circulating CD4^+^CD25^+^ (Treg) and CD3^+^CD56^+^ (T_R3-56_) regulatory T-cell subsets in CLL subjects with stable disease. Panel A to D show comparative analysis of the percentage and the number of circulating Treg and T_R3-56,_ in CLL subjects and healthy controls. Decreased percentage (**A**) and increased number (**B**) of the Treg cells has been shown to be associated with not significant difference in percentage of circulating T_R3-56_ lymphocytes (**C**) and increased number of this T-cell subset (**D**) in the CLL cohort. Grey and white columns indicate data obtained in healthy controls (CTR in *x* axis) and CLL individuals (CLL in *x* axis), respectively. Statistical evaluation of data has been performed by means of the *Mann–Whitney* test. Panel (**E**) shows the significant correlation, as evaluated by *Spearman’s* test, between percentage of circulating Treg and T_R3-56_ lymphocytes in CLL subjects. Statistical significance values are indicated. NS indicates the not statistically significant value. The applied flow cytometry gating strategy is reported in in Section 4.2.

**Figure 3 ijms-24-09596-f003:**
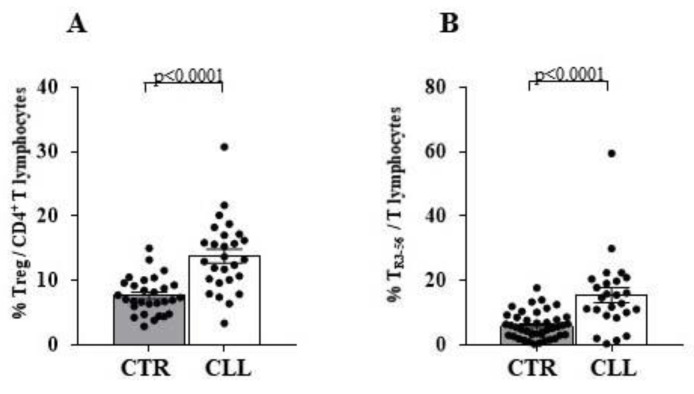
Percentage of Treg and T_R3-56_ lymphocytes are significantly increased in the T-cell compartment of CLL subjects. Panel (**A**) shows comparative analysis of the percentage of the Treg subset in the CD4^+^ T-cell population in the CLL subjects, as compared with controls. As shown, significant increase in Treg cells has been revealed in the CLL cohort; panel (**B**) shows percentage of T_R3-56_ lymphocytes in the T cells of CLL subjects, as compared with healthy individuals. As shown, significant increase in T_R3-56_ cells has been revealed in the CLL cohort. Grey and white columns indicate data obtained in healthy controls (CTR in *x* axis) and CLL individuals (CLL in *x* axis), respectively. Statistical evaluation of data has been performed by means of the *Mann–Whitney* test. Statistical significance values are reported. The applied flow cytometry gating strategy is reported in Section 4.2.

**Figure 4 ijms-24-09596-f004:**
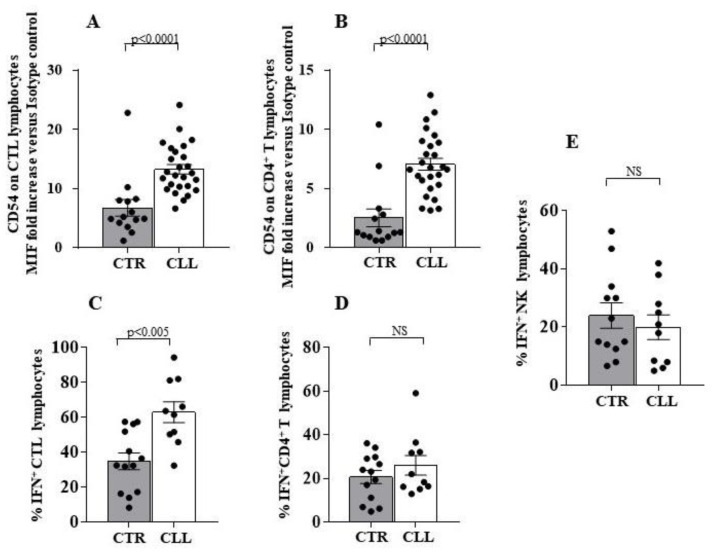
Significant increase in surface CD54 expression and higher interferon (IFN)-γ production characterises cytotoxic T cells of CLL subjects with stable disease. Panel (**A**,**B**) refer CD54 expression level on CD8 and CD4 T cells, respectively. As detailed in the Materials and Methods Section, immune-fluorescence data were expressed as ratio of the mean intensity fluorescence (MIF) value for the CD4 and CD8 cells and the control MIF value obtained after staining the same cell population with the isotype control mAb. As shown, CD4 and CD8 T-cell effectors are characterised by significant increase in surface CD54 molecule expression in the CLL cohort. Panels (**C**–**E**) show IFN-γ production by CD8, CD4 T lymphocytes and NK cells, respectively. Cytokine production analysis of has been performed by immune-fluorescence and flow cytometry detection after an over-night (ON) culture in the presence of PMA and ionomycin (see Materials and Methods Section for details). Grey and white columns indicate data obtained in healthy controls (CTR in *x* axis) and CLL individuals (CLL in *x* axis), respectively. For the comparative analysis of CD54 expression on T-cell effectors, immune-fluorescence data were expressed as ratio of mean intensity fluorescence (MIF) value for the CD4 and CD8 T-cell subset and the control MIF value obtained after staining the same cell population with the isotype control mAb. Statistical evaluation of data has been performed by means of the *Mann–Whitney* test. Statistical significance values are indicated. NS indicates the not statistically significant value. The applied flow cytometry gating strategy is reported in Section 4.2.

**Figure 5 ijms-24-09596-f005:**
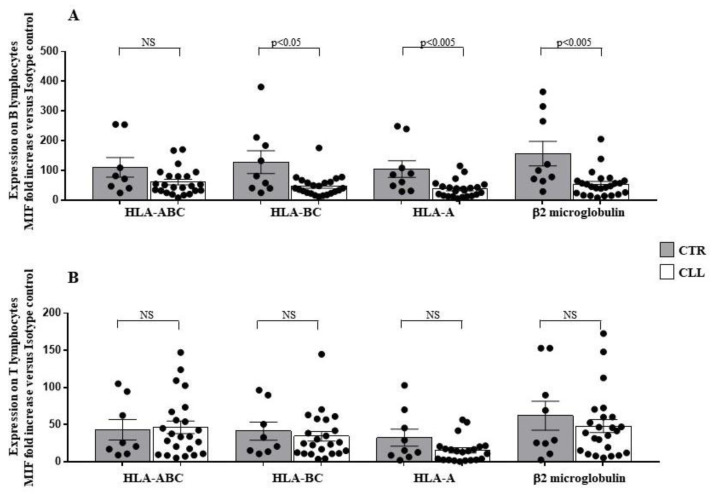
Significant decrease in human leukocyte antigen (HLA)-A and HLA-BC expression level characterises B lymphocytes of CLL subjects with stable disease. Panel (**A**,**B**) show HLA-ABC, HLA-A and HLA-BC surface expression level on B and T lymphocytes, as indicated. As shown, significant decrease in surface expression of HLA-A and HLA-BC molecules level has been observed in the B-cell compartment of the CLL subjects, as compared with healthy controls. No significant changes were revealed, for all the HLA-I molecules analysed, in the T lymphocytes of the CLL cohort. As detailed in the Materials and Methods Section, immune-fluorescence data were expressed as ratio of the mean intensity fluorescence (MIF) value for B and T cells and the control MIF value obtained after staining the same cell populations with the isotype control mAb, as described [52,55]. Grey and white columns indicate data obtained in healthy controls (CTR caption of (**A**,**B**) Panels) and CLL individuals (CLL caption of (**A**,**B**) Panels), respectively. Statistical evaluation of data has been performed by means of the *Mann–Whitney* test. Statistical significance values are indicated NS indicates the not statistically significant value. The applied flow cytometry gating strategy is reported in Section 4.2.

**Figure 6 ijms-24-09596-f006:**
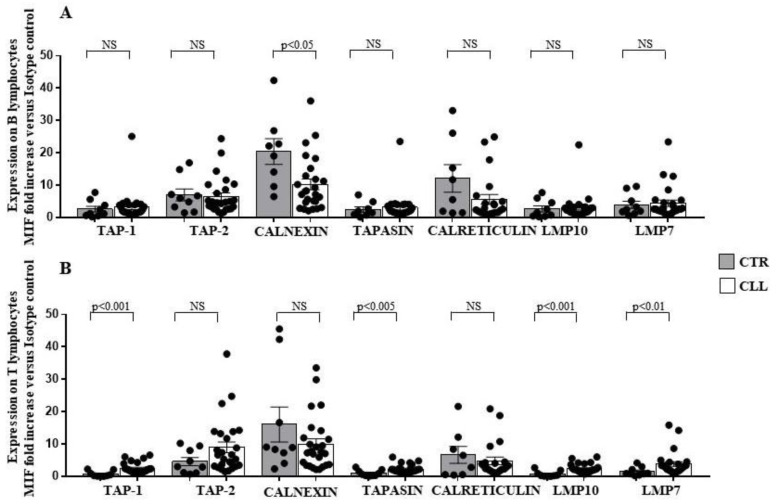
Significant decrease in intracellular calnexin expression in B lymphocytes, associated with up-regulation of TAP-1, Tapasin, LMP7 and LMP10 in the T-cell compartment, characterises CLL subjects with stable disease. Panel (**A**,**B**) show TAP-1, TAP-2, calnexin, tapasin, calreticulin, LMP10 and LMP7 intracellular expression in B- and T-cell compartment, as indicated. As shown, significant decrease in intracellular calnexin has been observed in the B cells of the CLL subjects, as compared with controls; at variance significant increase in TAP-1, Tapasin, LMP10 and LMP7 was revealed in the T lymphocytes of the CLL cohort. As detailed in the Materials and Methods Section, intracellular immune-fluorescence data were expressed as ratio of the mean intensity fluorescence (MIF) value for B and T cells and the control MIF value obtained after staining the same cell populations with the isotype control mAb. Grey and white columns indicate data obtained in healthy controls (CTR caption of (**A**,**B**) Panels) and CLL individuals (CLL caption of (**A**,**B**) Panels), respectively. Statistical evaluation of data has been performed by means of the *Mann–Whitney* test. Statistical significance values are indicated. NS indicates the not statistically significant value. The applied flow cytometry gating strategy is reported in Section 4.2.

**Figure 7 ijms-24-09596-f007:**
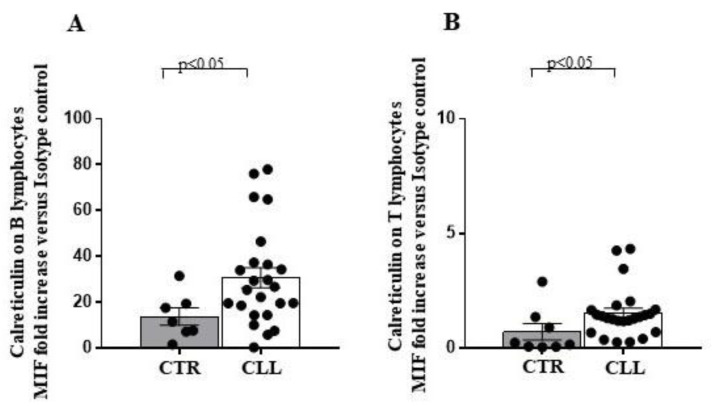
Higher surface calreticulin expression, as a measure of the overloading of the intracellular protein folding capacity, characterises circulating B and T lymphocytes of CLL subjects, as compared with healthy controls. Panel (**A**,**B**) show evaluation of surface calreticulin expression on circulating B and T lymphocytes of CLL individuals and healthy controls, as indicated. As shown, significant increase in the surface calreticulin expression has been revealed in both B and T cells from the CLL subjects. As detailed in the Materials and Methods Section, immune-fluorescence data were expressed as ratio of the mean intensity fluorescence (MIF) value for B and T cells and the control MIF value obtained after staining the same cell populations with the isotype control mAb. Grey and white columns indicate data obtained in healthy controls (CTR in *x* axis) and CLL individuals (CLL in *x* axis), respectively. Statistical evaluation of data has been performed by means of the *Mann–Whitney* test. Statistical significance values are indicated. The applied flow cytometry gating strategy is reported in in Section 4.2.

**Figure 8 ijms-24-09596-f008:**
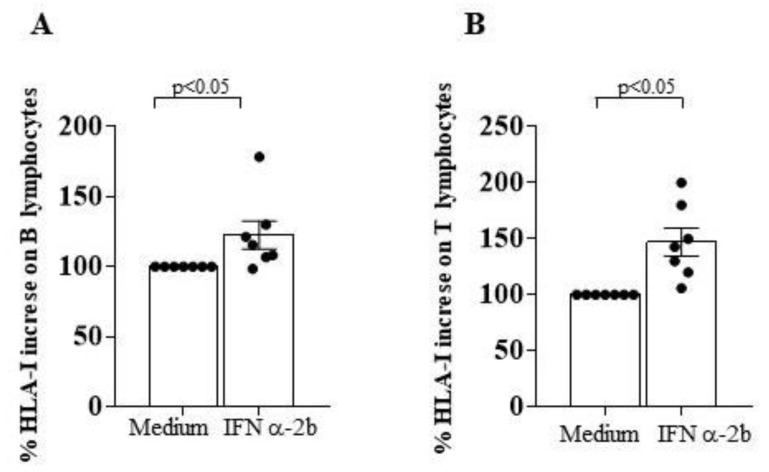
In vitro treatment with interferon (IFN) α-2b significantly increases HLA-I expression on B and T lymphocytes of CLL subjects with stable disease. Panel (**A**,**B**) indicate % increase in HLA-I surface expression level on B and T cells of CLL subjects after an overnight culture in the presence of medium alone or saturating concentration of IFN α-2b, as indicated in *x* axis. Statistical evaluation of data has been performed by means of the *Wilcoxon matched-pairs signed rank* test. Statistical significance values are indicated. The applied flow cytometry gating strategy is reported in Section 4.2.

**Figure 9 ijms-24-09596-f009:**
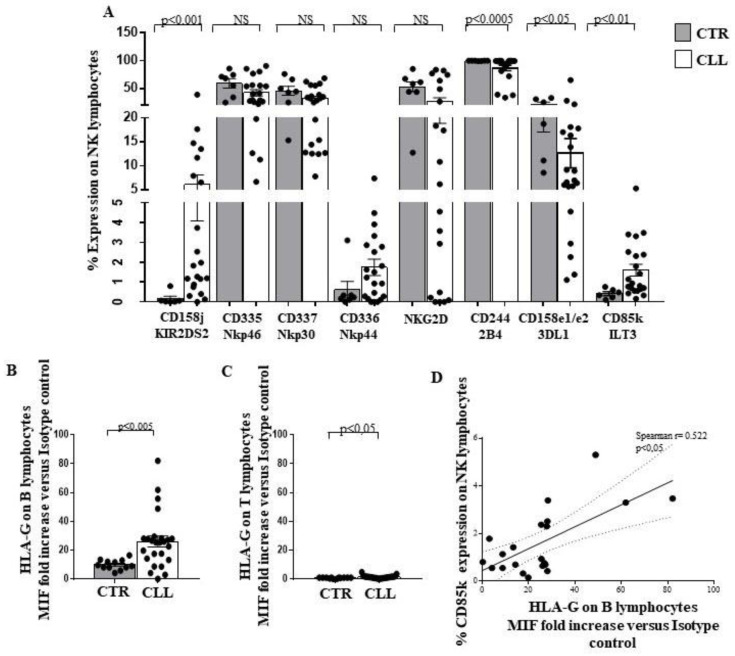
Analysis of activating and inhibiting receptor repertoire profile of circulating NK effectors of CLL subjects with stable disease, as compared with healthy controls. Panel (**A**) shows the expression of KIR2DS2, CD335, CD336, CD337, NKG2D, CD244, 3DL1 and CD85k on the surface of circulating NK effectors of CLL subjects, as compared with controls. As shown, significant increase in the activating KIR2DS2 receptor, associated with reduced level of the inhibiting structure 3DL1 has been observed together with a decrease in the activating CD244 molecule and an increase in the inhibiting CD85k receptor, able to bind the HLA-G molecule. Panel (**B**,**C**) show HLA-G expression level on the surface of B and T lymphocytes of the CLL cohort, as compared with controls. Significant increase of HLA-G expression has been revealed on both cell subsets in the CLL subjects. Grey and white columns indicate data obtained in healthy controls (CTR caption of Panel A and in *x*-axis of (**B**,**C**) Panels) and CLL individuals (CLL caption of Panel (**A**) and in *x*-axis of (**B**,**C**) Panels), respectively. Statistical evaluation of data has been performed by means of the *Mann–Whitney* test. Panel (**D**) shows the correlation analysis, as evaluated by the *Spearman’s* test, between the percentage of circulating NK cells expressing CD85k receptor and the HLA-G level on the B lymphocytes of the CLL subjects. Statistical significance values are indicated. NS indicates the not statistically significant value. The applied flow cytometry gating strategy is reported in Section 4.2.

**Figure 10 ijms-24-09596-f010:**
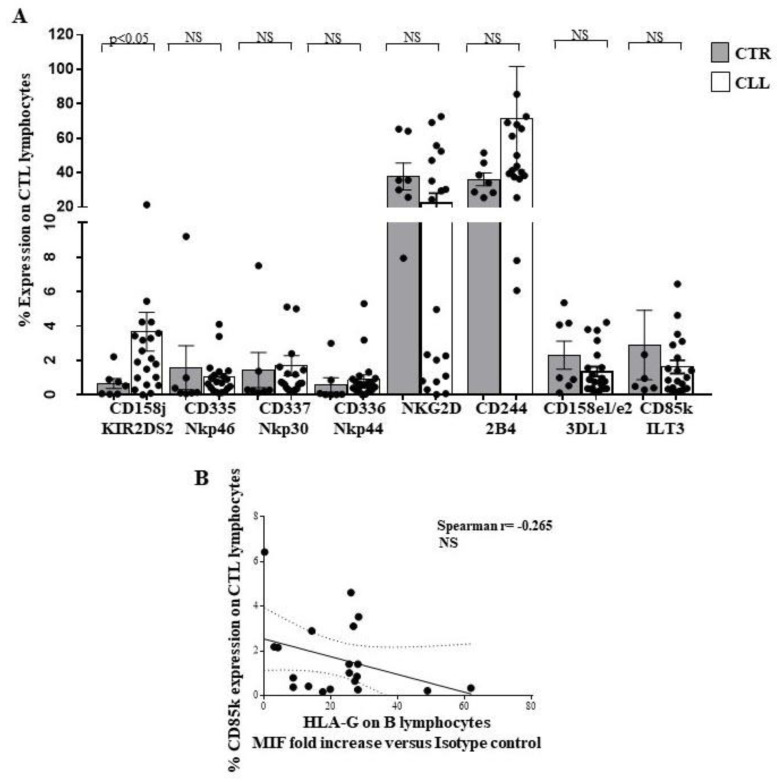
Analysis of receptor repertoire profile of circulating CTL effectors of CLL subjects with stable disease, as compared with healthy controls. Panel (**A**) shows the expression of KIR2DS2, CD335, CD336, CD337, NKG2D, CD244, 3DL1 and CD85k on the surface of circulating CD8^+^ T-cell effectors of CLL subjects, as compared with controls. As shown, significant increase in the activating KIR2DS2 receptor has been observed. Grey and white columns indicate data obtained in healthy controls (CTR caption of Panel (**A**)) and CLL individuals (CLL caption of Panel (**A**)), respectively. Statistical evaluation of data has been performed by means of the *Mann–Whitney* test. Panels (**B**) shows the correlation analysis, as evaluated by the *Spearman’s* test, between the percentage of circulating CD8^+^ T cells expressing CD85k receptor and the HLA-G level on the B lymphocytes of the CLL subjects with stable disease. Statistical significance values are indicated. NS indicates the not statistically significant value. The applied flow cytometry gating strategy is reported in Section 4.2.

**Figure 11 ijms-24-09596-f011:**
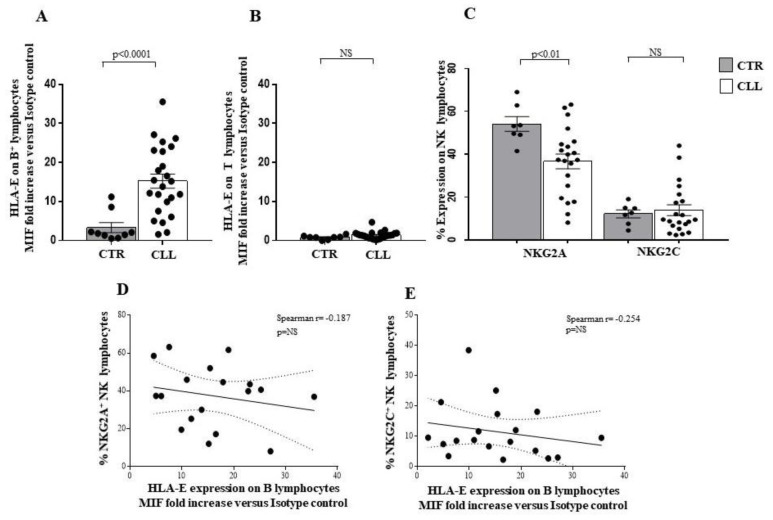
Analysis of HLA-E on B and T lymphocytes, and of NKG2A and NKG2C expression on circulating NK effectors of CLL subjects with stable disease. Panel (**A**,**B**) show the expression level of the HLA-E molecule on the surface of B and T lymphocytes, respectively. Significant increase in the HLA-E molecule on the surface of CLL B lymphocytes has been observed. Panel C shows the percentage of the HLA-E binding receptors NKG2A and NKG2C on circulating NK lymphocytes of the CLL cohort, as compared with controls. As shown, significant reduction of the inhibiting NKG2A receptor has been observed in the LLC cohort. Grey and white columns indicate data obtained in healthy controls (CTR in *x-axis* of A and B Panels and CTR caption in (**C**) Panel) and CLL individuals (CLL in *x*-axis of (**A**,**B**) Panels and CLL caption of C Panel) respectively. Statistical evaluation of data has been performed by means of the *Mann–Whitney* test. Panels (**D**,**E**) show the correlation analysis, as evaluated by the *Spearman’s* test, between the percentage of circulating NK cells expressing NKG2A or NKG2C receptors and the HLA-E expression level on the B lymphocytes of the CLL subjects with stable disease. Statistical significance values are indicated. NS indicates the not statistically significant value. The applied flow cytometry gating strategy is reported in Section 4.2.

**Figure 12 ijms-24-09596-f012:**
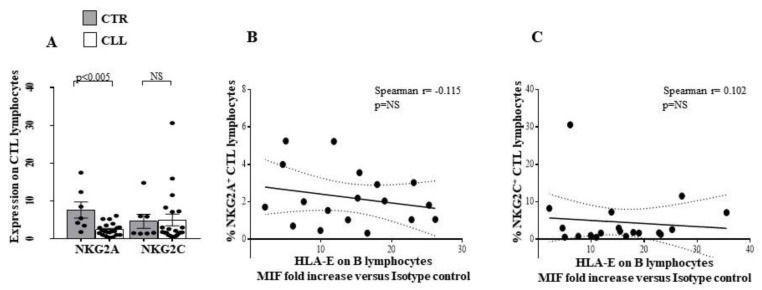
Comparative analysis of NKG2A and NKG2C receptor expression on circulating CTL effectors of CLL subjects with stable disease, as compared with healthy controls. Panel (**A**) shows the percentage of the HLA-E binding receptors NKG2A and NKG2C on circulating CD8^+^ T lymphocytes of the CLL subjects, as compared with controls. As shown, significant reduction in the inhibiting NKG2A receptor has been observed in the LLC cohort. Grey and white columns indicate data obtained in healthy controls (CTR caption of Panel (**A**)) and CLL individuals (CLL caption of Panel (**A**)), respectively. Statistical evaluation of data has been performed by means of the *Mann–Whitney* test. Panels (**B**,**C**) show the correlation analysis, as evaluated by the *Spearman’s* test, between the percentage of circulating CTL expressing NKG2A or NKG2C receptors and the HLA-E expression level on the B lymphocytes of the CLL subjects. Statistical significance values are indicated. NS indicates the not statistically significant value. The applied flow cytometry gating strategy is reported in Section 4.2. No differences were revealed between the level of NKG2A/NKG2C on CTL and the expression of HLA-E on B lymphocytes of CLL subjects (Panel (**B**,**C**), respectively). Taken together, these data indicate that an activation profile appears to characterise the NK effector recognition repertoire of CLL subjects with stable disease.

**Figure 13 ijms-24-09596-f013:**
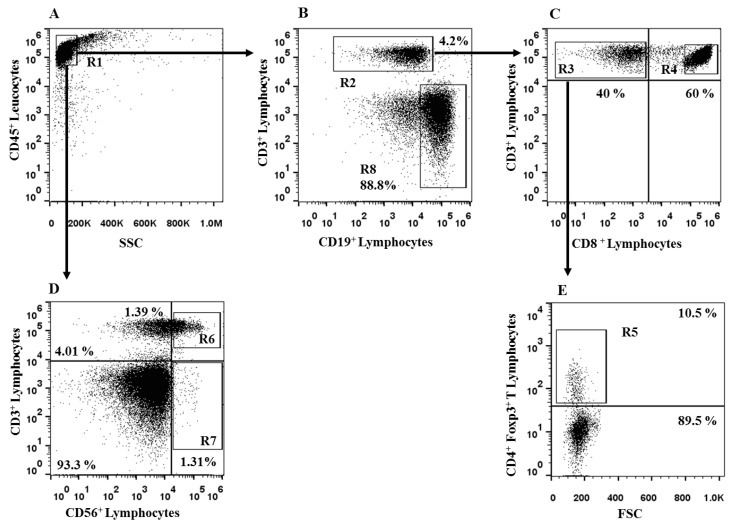
Flow cytometry gating strategy for analysis of CLL subjects with stable disease. Panel (**A**) shows the R1 gating on circulating CD45^+^ cells of the CLL subjects. From R1 region, we identified CD3^+^ T cells (R2, Panel (**B**)). From R2, we gated the CD3^+^ CD8^-^ (R3, Panel (**C**)) and CD3^+^ CD8^+^ T cells (R4, Panel (**C**)). From R3 region, we observed CD4^+^ Foxp3^+^ Treg (R5, Panel (**E**)). From R1 region (Panel (**A**)), we identified CD3^+^ CD56^+^ T_R3-56_ cells (R6, Panel (**D**)) and CD3^-^ CD56^+^ NK lymphocytes (R7, Panel (**D**)). From R1 gate, the B cells were identified by the CD19 expression (R8, Panel (**B**)). The figure reports one representative experiment performed in CCL subjects.

**Table 1 ijms-24-09596-t001:** Patient clinical and haematological characteristics.

Parameters	Mean ± SD	Range
Male/Female ratio 15:11		
Age (years)	68 ± 7.23	45–74
Duration of illness (years)	8.91 ± 4.46	3–18
Haemoglobin (g/dL)	13.79 ± 1.26	11.9–16.5
Platelet count (×10^9^/L)	188.2 ± 65.85	107–384
White blood cell count (×10^9^/L)	24.82 ± 22.92	5.86–109.2
Neutrophil count (×10^9^/L)	9.97 ± 8.59	0.61–34.94
Lymphocyte count (×10^9^/L)	15.97 ± 15.59	3.96–74.26

Patients did not show lymphadenopathy greater than 2 cm in all the superficial lymph-node stations; the splenic dimensions are stable over time, with a splenic volume ranging between 250 and 400 mL.

## Data Availability

Data supporting reported results can be obtained by corresponding authors (G.R. and G.T.).

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
