# Peer review of "Adaptive and Innate Cytotoxic Effectors in Chronic Lymphocytic Leukaemia (CLL) Subjects with Stable Disease"

_ijms, 2023, doi:10.3390/ijms24119596_

Round 1

Reviewer 1 Report

Rubine and colleagues present a manuscript with interesting data on the profile of cells/molecules involved in adaptive and innate immunity in CLL patients (compared to healthy donors). Data are robust, well presented, and adequately discussed.

Minor point: The manuscript is well written but would benefit from proofreading by a native English editor.

The manuscript is well written but would benefit from proofreading by a native English editor.

Author Response

Minor point: The manuscript is well written but would benefit from proofreading by a native English editor.

Response 1: The authors are very grateful to the Reviewer for the very positive evaluation of the manuscript. As suggested, a careful revision of the English was performed (see revisions made to the manuscript tha are marked up using the “Track Changes” function by using MS Word, as recommended by the Editor).

Reviewer 2 Report

The research group with substantial expertise in the field of tumor immunology has presented an impressive and lucid exposition of novel findings.

Minor comments:

·         Lines 90-92: as to my knowledge, the term TR3-56 is only used by your research group. However, this CD3+/CD56+ lymphocyte subset is a well-known cell population described as NKT-like cells in the literature. Please, include this in the Introduction. It is also known that the group is heterogenous regarding CD4/CD8 expression (Meggyes et al. 2023). Could there also be functional diversity?

·         Line 99: gating strategy should be demonstrated.

·         Lines 106-107: “counterbalanced by an increase in the absolute number of T…” – why is there a counterbalance? Please, explain….

·         Figures: please indicate on the X axis or the figures the investigated groups (grey-control, white-CLL)

·         Lines-463-464: “Overall, our data suggest that the presence of activated CTL might be involved in the control of the expansion of transformed B-cell clones” – the data do not fully support this statement since a comparison of the stable disease was only made with healthy controls. Please explain and briefly comment on why the study did not include advanced-stage CLL patients. A comparison of the data with this group would make the statement more reliable.

·         Lines 498-500:  “the increment of HLA-G and HLA-E on B cells represents a putative mechanism of escape from CTL and NK immune effectors recognition and elimination in CLL patients” – this statement is contrary to the previous one (Lines-463-464) questioning effective control mechanisms of activated CTLs in stable CLL.  Of course, the balance between them is crucial regarding disease outcomes. Please clarify.

·         Since this research is based on immunophenotyping and not on functional analysis, please make a short note on limitations of the study.

Author Response

General Reviewer’s Comment: The research group with substantial expertise in the field of tumor immunology has presented an impressive and lucid exposition of novel findings.

Response to General Comment: The authors thank the Reviewer for this very gratifying comment.

Minor comments:

Point 1: Lines 90-92: as to my knowledge, the term TR3-56 is only used by your research group. However, this CD3+/CD56+ lymphocyte subset is a well-known cell population described as NKT-like cells in the literature. Please, include this in the Introduction. It is also known that the group is heterogenous regarding CD4/CD8 expression (Meggyes et al. 2023). Could there also be functional diversity?

Response 1: Overall data generated from our laboratory indicated that TR3-56 cells represent a peculiar T cell subset phenotypically, metabolically and genetically distinct form NKT subset (Terrazzano et al. Nature metabolism 2021). In this regard, it is generally accepted that the NKT cells are a CD1d-restricted and a very rare (0.01-1% of PBL in humans) T cell subset (Godfrey, D.I. et al. Nat. Rev. Immunol. 2004; Lee, P.T. et al. J. Clin. Invest. 2002; Brennan, P.J. et al. Nat. Rev. Immunol. 2013; Pellicci DG, Uldrich AP Seminars in cell & development biology 2018; Pellicci DG et al. Nat. Rev. Immunol. 2020). NKT cells encompass two different subsets: the invariant NKT cells co-expressing a semi-invariant TCR (Valpha14 in mice and Valpha24/Vbeta11 in human) and a second one not expressing the semi-invariant TCR. In addition, it has been reported an additional "NKT like" cell subset able to kill K562 cell line, a prototypic NK target (Dons'koi, B.V. et al. J. Immunol. Methods 2011; Kuylensterna, C. et al. Eur. J. Immunol. 2011).

As by us described (Terrazzano et al. Nature metabolism xxxx), TR3-56 cells: (i) are not CD1d-restricted; (ii) do not express Valpha24/Vbeta11 chains but display a heterogeneous Vb repertoire; (iii) are unable to kill K562 cells in vitro; In addition, (iv) only 1-5% of CD1d-restricted T cells are positive for CD56 molecule, as showed in the Figure 1 included only for review consideration. Such population, by us associated with the ability to control CTL functional effectiveness, has been previously described as significantly decreased in Acute Myeloid leukaemia regression (Guo W, Xing C, Dong A, et al. Numbers and cytotoxicities of CD3+CD56+ T lymphocytes in peripheral blood of patients with acute myeloid leukaemia and acute lymphocytic leukaemia. Cancer Biol Ther. 2013; 14:916-921). Moreover, the significant increase of BM TR3-56 from very low/low risk to high/very high risk MDS subjects and the inverse correlation with the cytotoxic T-cell (CTL) activity has been recently confirmed (Serio et al Eur. J Haematol, 2023)

We also fully agree with the reviewer about the potential relevance of CD4/CD8 differential expression by the T cell population co-expressing CD3 and CD56 molecules, by us identified as the TR5-56 subset. In this context, we analysed their functional activity as a whole population (Terrazzano et al. Nature metabolism 2021) but studies are in progress to deeply investigate the functional role of both the CD8 or CD4 expressing TR5-56 lymphocytes. This observation, as requested, will be added in the discussion section of the revised manuscript (538-542 lanes of revised manuscript) together with the citation of the Meggyes M et al. paper showing the potential involvement of CD3+CD56+ lymphocytes in the immune-mediated processes underlying pre-eclampsia establishment.

Point 2: Line 99: gating strategy should be demonstrated.

Response 2: We agree with the Reviewer on the need to specify the flow cytometry gating strategy. In this regard, it has been reported in the revised manuscript (640-648 lanes) by adding the new figure 13 with an explanatory legend. We hope we have complied with the Reviewer's request.

Point 3: Lines 106-107: “counterbalanced by an increase in the absolute number of T…” – why is there a counterbalance? Please, explain….

Response 3: We agree with the reviewer that "counterbalanced " is confusing and does not correspond to clear and useful information. We apologize to the Reviewer and, in the revised version, we changed to "is accompanied" (16-16 lanes of revised manuscript).

Point 4: Figures: please indicate on the X axis or the figures the investigated groups (grey-control, white-CLL)

Response 4: We apologize to the reviewer: in the current version, all figures have the caption in x axis and/or in the panel.

Point 5: Lines-463-464: “Overall, our data suggest that the presence of activated CTL might be involved in the control of the expansion of transformed B-cell clones” – the data do not fully support this statement since a comparison of the stable disease was only made with healthy controls. Please explain and briefly comment on why the study did not include advanced-stage CLL patients. A comparison of the data with this group would make the statement more reliable.

Response 5: We agree with Reviewer concerning the relevance to extend the study on unstable and/or advances-stage of CLL patients. Indeed, the role of immune-dependent phenomena in the control of leukaemia clones in CLL is still largely controversial. Cytotoxic immune effectors and IFN-gamma dependent pathways have been largely associated with tumor control in human and mouse models (Schreiber, R.D. et al Science 2011; Mittal, D. et al. Curr Opin Immunol 2014). In this context, the possibility that the enrolment of a very homogeneous patient cohort, characterised by long-term stability of the transformed B cell clones, was by us considered as very relevant to properly investigate such issue.

However, it is undeniable that the study also extended to patients with unstable and/or advanced stage CLL could have benefited from more careful comparisons to allow for an understanding of the observations made. Therefore, in the revised manuscript, we have added the sentence "However, in order to demonstrate the relevance of this hypothesis in CLL subjects, further studies evidencing similarities and/or differences between subjects with stable and advanced-stage CLL regarding the functional role of both the CD8 or CD4 ex-pressing TR5-56 lymphocytes will be needed [50-51] (see also Study Limitations)" (538-542 lanes) and a "Study Limitations" paragraph (at the end of the manuscript) reporting the aforementioned limitation.

Point 6: Lines 498-500:“the increment of HLA-G and HLA-E on B cells represents a putative mechanism of escape from CTL and NK immune effectors recognition and elimination in CLL patients” – this statement is contrary to the previous one (Lines-463-464) questioning effective control mechanisms of activated CTLs in stable CLL.  Of course, the balance between them is crucial regarding disease outcomes. Please clarify.

Response 6: HLA-E and HLA-G expression by tumor clones represents a potential immune-escape mechanisms, especially for innate cytotoxic effectors. Moreover, the expression of specific HLA-E and HLA-G receptors by CTL indicate their potential involvement, as accessory signals, also in adaptive cell regulatory networks. Our results indicate significant down-modulation of the inhibiting NKG2A molecule in both adaptive and innate cells, while for CD85k, the inhibiting HLA-G binding molecule, a different behaviour might be identified in NK (significant increase) respect to T cells (no significant change in the expression rate). In this context, the observed substantial prevalence of activating signals in innate and adaptive cytotoxic effectors need to be considered as a valuable criterion to properly investigate immune-mediated tumor progression control in LLC. However, we agree with the reviewer about the statement that, as a whole, our study mainly provides immune-phenotype analysis of innate and adaptive cytotoxic effectors.

Therefore, in the revised manuscript, we deleted the sentence “the increment of HLA-G and HLA-E on B cells represents a putative mechanism of escape from CTL and NK immune effectors recognition and elimination in CLL patients” and added the sentence "Alterations of HLA class I molecules, as well as HLA-G and HLA-E, on B cells are of potential interest, as they could represent a putative escape mechanism of neoplastic clone B from CTL and NK recognition in CLL patients with stable disease. However, since the balance between the control of CTL and NK effectors may be crucial for disease determinism and outcomes, further studies of the functional mechanisms of the immune response and its regulation in CLL disease are needed (see Study Limitations)" (579 -584 lanes) and a paragraph "Study Limitations" (at the end of the manuscript) showing the aforementioned limit.

Point 7: Since this research is based on immunophenotyping and not on functional analysis, please make a short note on limitations of the study.

Response 7: In the revised manuscript, we have added a "Study Limitations" paragraph (at the end of the manuscript) reporting the above limitation.

All the revision of the manuscript are marked up using the “Track Changes” function by using MS Word, as recommended by the Editor.
